# Simultaneous Noninvasive Detection and Therapy of Atherosclerosis Using HDL Coated Gold Nanorods

**DOI:** 10.3390/diagnostics12030577

**Published:** 2022-02-23

**Authors:** Rinat Ankri, Dorit Leshem-Lev, Hamootal Duadi, Emanuel Harari, Menachem Motiei, Edith Hochhauser, Eli I. Lev, Dror Fixler

**Affiliations:** 1Department of Physics, Ariel University, Ariel 4076414, Israel; 2Cardiac Research Labs, The Felsenstein Medical Research Center, Petah-Tikva 4922297, Israel; doritl2@clalit.org.il (D.L.-L.); hochhaus@post.tau.ac.il (E.H.); 3Faculty of Engineering and Institute of Nanotechnology and Advanced Materials, Bar-Ilan University, Ramat-Gan 5290002, Israel; hamootal@gmail.com (H.D.); menachem.motiey@biu.ac.il (M.M.); dror.fixler@biu.ac.il (D.F.); 4Cardiology Division, Assuta Ashdod University Hospital, Ben Gurion University of the Negev, 7 Harefu’a St., Ashdod 7747629, Israel; eharari@gmail.com (E.H.); elil@clalit.org.il (E.I.L.)

**Keywords:** noninvasive imaging, diffusion reflection, atherosclerosis, unstable plaques, gold nanorods, high-density lipoprotein

## Abstract

Cardiovascular disease (CVD) is a major cause of death and disability worldwide. A real need exists in the development of new, improved therapeutic methods for treating CVD, while major advances in nanotechnology have opened new avenues in this field. In this paper, we report the use of gold nanoparticles (GNPs) coated with high-density lipoprotein (HDL) (GNP-HDL) for the simultaneous detection and therapy of unstable plaques. Based on the well-known HDL cardiovascular protection, by promoting the reverse cholesterol transport (RCT), injured rat carotids, as a model for unstable plaques, were injected with the GNP-HDL. Noninvasive detection of the plaques 24 h post the GNP injection was enabled using the diffusion reflection (DR) method, indicating that the GNP-HDL particles had accumulated in the injured site. Pathology and noninvasive CT measurements proved the recovery of the injured artery treated with the GNP-HDL. The DR of the GNP-HDL presented a simple and highly sensitive method at a low cost, resulting in simultaneous specific unstable plaque diagnosis and recovery.

## 1. Introduction

Despite recent advances, atherosclerosis (AS) vascular disorder (ASVD) and its major vascular complications—myocardial infarction and ischemic cerebrovascular accident—remain the leading cause of morbidity and mortality in developed countries. Over the last decades, various therapeutic approaches to ASVD treatment have been developed, such as pharmacological treatments [1] and interventional techniques [2]. Oxidative stress and the formation of macrophage foam cells contribute to the initiation, progression and rupture of lipid-rich vascular lesions. Anti-inflammatory therapy methods were developed to treat ASVD [3], but most anti-inflammatory therapeutic agents delay the progression of ASVD rather than prevent its formation [4]. Therapeutic strategies that target early atherosclerotic lesions and enable plaque modification (by directly treating the underlying inflammation) may have a promising role in ASVD treatment [5]. Thus, a well-known therapeutic target for ASVD is the high-density lipoprotein (HDL) [6,7]. HDL exerts cardiovascular protection by promoting reverse cholesterol transport (RCT) and other pleiotropic and anti-inflammatory beneficial effects [6,7,8]. Previous studies have shown that HDL inhibits the chemotaxis of monocytes and adhesion of leukocytes to the endothelium, endothelial dysfunction and apoptosis. In addition, HDL inhibits low-density lipoprotein (LDL) oxidation, platelet activation and factor X activation, and also stimulates the proliferation of endothelial cells [9]. In this work, we used the anti-inflammatory effect of HDL to treat one of the pro-inflammatory lipoprotein-associated factors: phospholipase A2 (Lp-PLA2) [10]. Lp-PLA2 plays a key role in the development of atherosclerotic lesions and formation of a necrotic core, leading to a plaque which is highly vulnerable. Epidemiologic studies demonstrated that increased circulating levels of Lp-PLA2 predicted an increased risk of myocardial infarction, stroke and cardiovascular mortality [11,12,13]. HDL has been shown to inhibit Lp-PLA2 expression and activity in cell culture macrophages [10], and was suggested to serve as a natural agent that can be modified for biomedical imaging and therapeutic purposes [14,15,16].

In addition to the efforts being done in the development of a treatments for ASVD, its early detection is also important for enhancing the chances of a full recovery. ASVD remains asymptomatic for many years, and the identification of early-stage inflamed, unstable lesions within the coronary circulation is elusive, due to the small plaque size, cardiac and respiratory motion, and lack of a suitable marker specific to the unstable plaque [17,18,19]. Current commercially available ASVD detection methods include intravascular ultrasound (IVUS) and intravascular optical coherence tomography (IVOCT) [20]. The latter delivers a higher resolution and higher image contrast, but comes at the cost of the necessity to flush blood from the artery. This step, combined with the inherently limited ranging depth, make IVOCT less reliable in large vessels. IVUS is easier to use and more versatile, but the images it produces contain fewer details and lower contrast. In addition, both IVUS and IVOCT technologies do not provide information on the tissue composition on real-time, which has to be inferred by image interpretation, frequently giving rise to differing opinions. Near-infrared fluorescence (NIRF) has also been applied to add chemical tissue information to structural imaging for the detection of unstable AS. Similarly, fluorescence lifetime spectroscopy has been suggested to generate label-free optical molecular contrast, which may be useful for detecting critical atherosclerotic plaque. Still, despite these methods’ relatively high detection accuracy, their clinical value remains controversial [21,22].

In this study, we suggest the use of HDL bio-conjugated gold nanoparticles (GNPs-HDL) for noninvasive in vivo AS detection. GNRs are well-known contrast agents in biomedical imaging, showing high performance in optical coherence tomography (OCT) [23,24], magnetic resonance imaging (MRI) [25], coherence tomography (CT) [26,27], terahertz spectroscopy [28], and other imaging methods. All these methods are robust and widely used in clinical and/or research applications; however, they require sophisticated equipment, and do not provide a way to distinguish between stable and unstable AS plaques. In this work, GNPs-HDL were used for the direct identification of unstable AS plaques with diffusion reflection (DR) measurements—a simple and unique method for tracking unstable atherosclerotic plaques. In our previous work, we demonstrated that macrophages, which are a major component of unstable “vulnerable” atherosclerotic plaques [29,30], were able to take up GNPs in in vitro culture conditions, resulting in a change in their optical properties [31]. Moreover, we showed in vivo that a GNP-based DR method can clearly detect accumulations of macrophages in injured vascular tissue [31]. Based on the therapeutic potential of HDL as an inhibitor of Lp-PLA2, as well as the ability of macrophages to take up GNPs and be detected by diffusion reflection (DR) measurements, we report here the use of GNPs-HDL for the simultaneous detection and specific therapy of macrophage-rich vascular plaques.

## 2. Materials and Methods

The study protocol was approved by the Helsinki Committee of Rabin Medical Center and Sackler Faculty of Medicine, Tel Aviv University. Informed consent for participation in the study was obtained from participants. All methods were performed in accordance with the relevant guidelines and regulations of the protocol described above.

In vitro study: Macrophage cell culture, *n* = 10. Primary human macrophage cell cultures were incubated for 48 h with: (1) no additions; (2) GNPs; or (3) GNPs coated with HDL. GNP-HDL uptake was measured by inverted light microscopy.

In vivo study: Rat carotid artery balloon injury model. We chose this vascular injury model as an atherosclerotic model for the following reasons. First, similar to the unstable atherosclerotic plaque, macrophages and other mono-nuclear cells are recruited and infiltrate the arterial vessel wall following injury. Second, this process is rapid and enables the location of the macrophage-rich atherosclerotic plaque and facilitates precise monitoring [32,33].

Two weeks later, the rats were exposed to carotid balloon injury, and GNPs with or without a HDL coating were injected. Diffusion reflection (DR) measurement was taken 24 h after injection, and the influence of the HDL on the injured arteries was pathologically tested two weeks later.

### 2.1. Macrophage Cell Culture

Human peripheral blood mononuclear cells (PBMCs) were isolated from healthy blood donors by density-gradient centrifugation on Ficoll-Hypaque as previously described [31]. Briefly, PBMCs (106 cells mL) were first seeded into 24-well plates (0.5 mL per well). After 2 h, non-adherent cells were removed by several washes with warm PBS. Freshly isolated monocytes were allowed to differentiate into macrophages in complete RPMI1640 with or without human recombinant macrophage colony stimulating factor (100 ng/mL) for 6 days. Macrophage cell culture viability was assessed by using an MTT viable test kit (Sigma-Aldrich, St. Louis, MO, USA) according to the manufacturer’s protocol.

To confirm macrophage cell lineage, direct immunostaining was performed with antibodies directed against cd11b/mac1 (Biolegend Inc., San Diego, CA, USA). Eighty percent of positive staining, compared to isotype control, was observed by both flow cytometry analysis and by inverted microscopic analysis (results not shown, see Ankri et al., 2014) [31].

### 2.2. Fabrication and Injection of Gold Nanoparticles

Two types of GNPs were utilized in this study: gold nanospheres (GNSs) and gold nanorods (GNRs). Thus, 30 nm GNSs were prepared using sodium citrate according to the methodology described by Enüstun and Turkevich [34]. The GNSs were coated with a PEG layer consisting of a mixture of thiol-polyethylene-glycol (mPEG-SH) (~85%, MW ~5 kDa) and a heterofunctional thiol–PEG–acid (SH–PEG–COOH) (~15%, MW ~3.4 kDa) (Creative PEGWorks, Winston Salem, NC, USA), and were then attached to rHDL (Sigma-Aldrich, Rehovot, Israel). The particles were also characterized using transmission electron microscopy (TEM), zeta potential measurements and dynamic light scattering (DLS).

The GNRs were synthesized using the seed-mediated growth method [35]. A solution of GNRs suspended in cetyltrimethylammonium bromide (CTAB) (Sigma-Aldrich, St. Louis, MO, USA) was centrifuged at 11,000× *g* for 10 min, decanted and resuspended in water to remove excess CTAB. In order to prevent aggregation, and to stabilize the particles in physiological solutions, a layer of polyethylene glycol (mPEG–SH, MW 5000 g/mol) (creative PEGWorks, Winston Salem, NC, USA) was adsorbed onto the GNRs. A 200 μL mixture of mPEG–SH (5 mM) (85%) and SH–PEG–COOH (1 mM) (15%) was added to 1 ml of GNRs solution. The mixture was stirred for 24 h at room temperature, then PEGylated GNRs were covalently conjugated with HDL (Sigma-Aldrich, Rehovot, Israel). The GNRs were also characterized using transmission electron microscopy (TEM), zeta potential measurements and DLS.

200 µL of GNRs and 10 mg/mL of GNR-HDL were injected into the rats’ carotids after their necks were shaved to enable the irradiated light to penetrate the carotid.

### 2.3. LPPLA2 Level

LPLA2 levels were measured on macrophage cell culture media that were collected using Lp-pla2 ELISA assay, according to the manufacturer’s instructions (human lipoprotein-associated phospholipase A2, Lp-PL-A2 ELISA Kit, Sunlong Biotech Co., Hangzhou, China).

### 2.4. A Rat Carotid Artery Balloon Injury Model

Adult male Wistar rats (Charles River, MA, USA) weighing 400 g were anesthetized with pentobarbital (50 mg/kg, intraperitoneally) to cause a balloon injury in the rat carotid artery in the adult male as previously described [31]. Carotids were collected 7, 14 and 28 days post injury (after intraperitoneal injection of a lethal dose of pentobarbital), placed in a cryomatrix (OCT) and frozen at −80 °C. In addition, 12 μm cross-sections were made from the entire length of the carotid and were used for immunohistochemistry analysis.

### 2.5. Diffusion Reflection Measurements

A noninvasive optical technique was designed and built (Negoh-Op Technologies, Yehud, Israel) for DR measurements, as was previously described [36]. Briefly, the set-up included a laser diode with wavelengths of 650 and 780 nm as an excitation source. Irradiation was carried out using a 125 μm diameter optic fiber to achieve a pencil beam illumination. The irradiation was only 5 mV in intensity, thus no heating effect was expected or observed. We used a portable photodiode that was kept in close contact with the tissue surface to prevent ambient light from entering the detection system and to avoid potential light loss through specimen edges. The initial distance ρ between the light source and the photodiode was ~1 mm. A consecutive reflected light intensity (Γ) measurement was enabled using a micrometer plate that was attached to the optical fiber. The micrometer plate was moved by incremental steps of 250 µm each, and the reflected light intensity was collected from 20 source-detector distances with ρ varying between 1 mm and 5 mm. The reflected intensity Γ(ρ), presenting units of volts per mm, was collected using a digital scope (Agilent Technologies, Mso7034a, Santa Clara, CA, USA), and data were processed using the LabView (National Instruments, Austin, TX, USA) program.

### 2.6. CT Measurements

All scans were performed using a micro-CT scanner (SkyScan, high-resolution model 1176). In vivo scans were performed at a nominal resolution of 8.5 microns employing an applied X-ray tube voltage of 50 kV, source current of 500 µA and 0.5 mm aluminum (Al) filtering. Ex vivo (artery) measurements were performed with the following scanning parameters: source voltage of 40 kVe, source current of 600 µA, pixel size of 12.12 µm and no filtering. Volume-rendered 3D images were generated using an RGBA transfer function 16 in SkyScan CT-Voxel (“CTVox”) software.

### 2.7. Statistical Analysis

Data are presented as means ± standard deviation. Comparisons of continuous variables were performed by Student’s t-test, and statistical significance was set at *p* < 0.05.

## 3. Results

### 3.1. Preparation and Characterization of GNSs and GNS-HDL

The successful attachment of the Au nanoparticles to the HDL was first verified. The synthesis of the nano-hybrid HDL coated GNP particles was based on the expected bonds between the amine group in the HDL and the carboxylic group of the PEG, which coated the GNPs as illustrated in Figure 1a. All in vitro measurements were performed with gold nanospheres (GNSs), and the in vivo measurements were performed with gold nanorods (GNRs), which presented an absorption peak at 650 nm (while the GNSs presented a peak at 530 nm; see Figure 1c), to enable deeper tissue penetration. Fourier transform infrared spectroscopy (FT-IR) measurements were performed. Figure 1b presents the FTIR spectra of GNSs and HDL coated GNSs. The FT-IR spectra presented major peaks around 1100 cm^−1^, related to the C–C bonds in the HDL and PEG molecules, and at 1700 cm^−1^ and 3400 cm^−1^, ascribed to C=O and O–H bonds in the PEG molecule. The band B at 1400 cm^−1^ is attributed to the COO^−^ symmetric stretching vibration from Asp and Glu residues [37]. This band is very strong in HDL, therefore it presents major evidence for the HDL presence in the GNS-HDL solution. The small peaks at 820 cm^−1^ and the major band denoted by A, at 1548 cm^−1^, present evidence of an amide band [38] resulting from the interaction between the carbon of the PEG molecule and the amine group of the HDL. In order to further confirm the attachment of the HDL to the gold nanoparticles, dynamic light scattering (DLS) and zeta potentials of the GNSs and the GNS-HDLs were performed; these are presented in Table 1. The DLS results show a clear increase in the particles’ radii following their HDL coating, which is additional proof of the successful binding of the HDL to the GNSs. The zeta potential of the GNS-HDLs was different from the potential of the bare GNSs, indicating the attachment of the protein to the GNSs. As the optical properties of the new hybrid nanoparticle are of high importance in this work, Figure 1c presents the absorption spectra of the GNSs and GNS-HDLs. The absorption peak remained the same, and the only observed change was the enlargement of the spectrum, resulting from the increase in the particle radii in the solution. Overall, Figure 1 presents a well-established proof of the binding of the HDL to the GNSs, and these results are relevant to all Au–HDL bonds, such as for the GNRs that were used for the in vivo measurements.

### 3.2. In Vitro Study of GNSs and GNS-HDL Uptake by Macrophages

The GNS-HDL uptake by the macrophages was then examined in vitro. Primary human macrophage cell cultures (*n* = 10) were incubated with HDL or GNSs coated HDL (GNS-HDL, 10 mg/mL; the final gold concentration in the cell culture was 0.1 mg/mL) for 48 h at 37 °C. After incubation, the cells were analyzed using hyperspectral microscopy. Figure 2a shows microscopy images of macrophage cell cultures without GNSs and with GNSs and GNS-HDLs. The cellular uptake of the GNSs and GNS-HDL particles is visible as dark dots within the cells. Similar to results from our previous study, the macrophages with the GNPs show a high viability and present a healthy cell morphology. Figure 2b presents the reflectance spectra of the macrophages before and after their incubation with GNSs and GNS-HDLs. Reflectance peaks at 530 nm correlated with the absorption and reflectance peak of the GNSs (Figure 1c), indicating that there was GNSs uptake by the cells. The reflectance peak at 530 nm showed a higher intensity following the GNS-HDL uptake, compared with the GNSs uptake. Average reflectance intensities are shown in Figure 2c, presenting values of 11.8 and 14 intensity units for the GNSs and GNS-HDL uptake, respectively. Cell viability was also evaluated using an MTT test (Figure 2e), indicating that the HDL was not toxic to the cells and did not have a negative effect on the macrophages’ viability.

Once the GNS-HDL uptake by the macrophages was ensured, the effect of HDL on the LP-PLA-2 levels was measured (Figure 2d). Macrophage cell culture media were collected and the LP-PLA-2 levels were determined using an LP-PLA-2 ELISA assay. Macrophage culture with GNS-HDL was associated with a significant reduction in LP-PLA-2 levels compared with the control (macrophages without GNSs) or the culture with GNSs only: control and cells with GNSs presented LP-PLA-2 mean levels of 0.0134 ± 0.001 and 0.0133 ± 0.002 pg/mL (*p* = 0.4), respectively, while the macrophage cell culture incubated with GNS-HDL presented a significantly lower LP-PLA-2 level of 0.0125 ± 0.001 pg/mL (*p* = 0.01).

### 3.3. In Vivo GNRs Accumulation in Injured Arteries Ensured with Noninvasive DR Measurements

Once the therapeutic effect of the GNS-HDL on the macrophages was ensured, in vivo studies of the theranostic effect of gold nanorods conjugated with HDL (GNR-HDL) were performed. The GNRs were chosen as contrast agents for the in vivo experiments, as they presented the strongest absorption properties, rather than GNSs. Figure 3 presents an illustrative description of the gold nanorod conjugation with HDL (Figure 3a), as well as the GNRs optical properties and TEM image (Figure 3b,c, respectively). The zeta potentials and DLS measurements of GNRs and GNR-HDLs are presented in Table 2, which suggest the attachment of the HDL to the GNRs.

The GNRs accumulation in the injured arteries was ensured by the noninvasive scanning of the rat neck using our DR system (see Section 2, Materials and Methods). As was previously described, the diffusion reflectance profile of an irradiated tissue depends on its absorption and scattering coefficients [39]. Once the GNRs accumulate in a tissue, the DR profile immediately changes due to the absorption properties of the Au nanoparticles [40,41]. The unstable plaque in the rats’ arteries was modeled by balloon injured carotid arteries [42]. In this work, we used the DR method to ensure the GNRs and GNR-HDL accumulations in the injured arteries 24 h post the GNRs injection. We irradiated the arteries with 650 nm illumination, which correlates with the injected GNRs absorption peak (see Figure 3b). GNRs and GNR-HDLs (directly injected into the arteries, 200 µL of GNRs, 10 mg/mL of GNR-HDL) were introduced into the injured arteries of two groups of rats, named *rat-1* and *rat-2*. Reflected diffusive light was then measured for both groups; representative reflectance spectra are presented in Figure 4. The DR profiles presented accumulations of GNRs, with and without HDL, in the injured arteries with increases of 0.27 (Figure 4b) and 0.21 (Figure 4c). These results are highly important, as they ensured, noninvasively, the accumulation of the GNRs within the injured arteries, enabling the leaving of HDL in the arteries in order to perform its curing effect. Thus, the therapeutic influence of the GNR-HDL on the injured arteries was tested two weeks later.

The rats’ carotids were extracted two weeks after the GNRs injection to test the injury condition following the HDL treatment. Figure 5 presents high-resolution CT ex vivo images of the injured rats’ arteries. The arteries were distorted due to the balloon injury. GNRs along the arteries can be clearly identified, marked as golden regions, since gold induces stronger X-ray attenuation [43]. The presented CT images demonstrate a clear difference between the artery that was treated with GNR-HDL and the artery that was treated with GNRs only: while gold still accumulated in the injured artery that was treated with GNRs only (Figure 5c), gold was almost entirely absent from the artery that was treated with GNR-HDL (Figure 5b). Since gold accumulation in the injured artery is attributed to GNRs uptake by macrophages and other mono-nuclear cells, the fact that no gold remained in the artery suggests that macrophages were no longer present within it in a significant amount at the time of imaging. Combined with our in vivo DR results, which showed 24 h post their injection that GNRs had accumulated in both arteries, these ex vivo CT images suggest the GNR-HDL nanohybrids had a therapeutic effect on the injured artery, as they attenuated the inflammatory response and macrophages were detached from the injured site.

### 3.4. Ex Vivo Validation of the Therapeutic Effect of the GNR-HDL

For further validation of the therapeutic effect of the GNR-HDL, pathological tests were performed. Two weeks following the GNRs injection (with or without HDL) the healthy and injured rats’ carotids were isolated and analyzed by hematoxylin and eosin (H and E) staining (Figure 5b,c) and by immunostaining with the macrophage cell marker CD 68 (Figure 5d,e). Figure 5a shows H and E staining of a healthy carotid, presenting normal anatomical structures of blood vessels without neointima tissue. Figure 5b,c present pathological results for the injured arteries 2 weeks post the GNR-HDL and GNRs injections, respectively. It can be clearly seen that the GNR-HDL treatment improved the artery injury, as the cellular-rich neointima reduced after the HDL injection. The CD 68 immunostaining demonstrated the absence of macrophages after the GNR-HDL treatment (Figure 5d), compared to a large accumulation of macrophages in the injured site as seen in Figure 5e. These important results prove that the unique GNR-HDL complex cured the injured site by decreasing the macrophage accumulation in the injured artery.

## 4. Discussion

Nanoparticles (NPs) represent a novel strategy for the diagnosis and treatment of AS, while new multifunctional nanoparticles with combined diagnostic and therapeutic capacities are under intensive investigation for AS theranostics [44,45,46]. Among other NPs, GNPs have unique properties that are exploited in the improvement of some of the techniques currently available for diagnosis, such as MRI, CT and PET [16,47,48]. Thus, e.g., Cormode et al. [49] imaged the vasculature in the arteries of atherosclerotic mice, showing the capability of the spectral CT system to detect the accumulation of Au-HDL in the aorta. This result was an indirect evaluation of macrophages in the plaques, and it was confirmed by TEM and confocal microscopy that macrophages incorporated the Au-HDL nanoparticles. Similarly, using Au-HDL nanoparticles, the assessment of plaques by CT and spectral CT was investigated as a means of characterizing the macrophage burden, calcification and plaque stenosis in an apolipoprotein E knockout (apo E-KO) mouse model of atherosclerosis [49]. GNPs were also used, combined with iodine-based contrast material (which was injected 24 h later) to be imaged by MRI, while macrophage targeting by Au-HDL was further evaluated by using transmission electron microscopy and confocal microscopy of aorta sections [50]. However, although these imaging methods are robust and provide relatively high-resolution imaging of AS plaques, they are highly expensive and complex, and use ionizing irradiance for their operation.

In the current study, we introduced a theranostics application of GNPs-HDL in ASVD characterized by macrophage infiltration, such as in unstable plaque [29,30], based on the simple and highly sensitive optical diffusion reflection method. GNPs have been originally applied in cancer detection and imaging, but their potential use in the imaging of cardiovascular diseases has drawn much attention from the research community in this field. The optical properties of GNPs have led to their utilization as contrast agents for optical or X-ray imaging modalities, allowing the detection of atherosclerotic plaques, intravascular thrombus and fibrotic tissues [51]. In our previous study, we showed that the diffusion reflection method enabled the detection of macrophage accumulation following vascular injury [31], which paved the way for the development of a DR-GNR novel detection tool for the identification of ASVD at its early stages, specifically for unstable macrophage-rich atherosclerotic plaques. In this study, we attempted to employ GNPs as drug carriers in the setting up of an atherosclerotic vascular injury model by coating them with HDL, and to track their therapeutic effect using the DR method.

GNRs coated with HDL molecules enabled the “access” of HDL to unstable atherosclerotic plaques, to perform its curing effect. A similar work was published a few years ago by Cormode et al., showing that gold-core HDL could be used for imaging, which enabled abdominal aorta of mice to be scanned using a small animal MRI scanner [16]. In addition, in 2008, Mulder and colleagues reported the development of gold-, iron (II) oxide- and quantum-dot-encapsulated HDL mimicking NPs for use in either computer tomography (CT) or MRI using conventional multistep methods [16]. In contrast to these works, we did not synthesize new nanoparticles, but instead attached the HDL to the well-known GNRs [52]. In vitro experiments showed the macrophage uptake of the GNS-HDL particles, (Figure 2a,b) demonstrating a slight superiority in the GNS-HDL uptake in comparison to that of GNSs only (Figure 2c). MTT tests revealed that the GNSs and GNS-HDLs did not affect the viability of the macrophages (Figure 2e). Then, the curing effect of the GNS-HDLs on the macrophages was tested by measuring the Lp-PLA2 levels in the macrophage plate (Figure 2d). Results revealed that the GNS-HDLs were indeed associated with lower Lp-PLA2 levels in the culture media compared to GNSs alone.

In the in vivo model, based on a rat carotid artery balloon injury, we showed the high efficiency of the DR method in the detection of the injured artery 24 h after GNRs and GNR-HDL injection. The results suggest the accumulation of both GNRs and GNR-HDL nanoparticles in the injured arteries after 24 h. The DR method enabled the noninvasive in vivo detection of GNRs’ accumulation in the injured artery close to the time of their injection. This noninvasive detection method enabled us to keep the rats alive in order to enable the HDL to affect the carotid and to perform its curing effect. Pathological examinations were performed two weeks later to reveal whether GNR-HDLs had reduced the inflammatory response in the injured artery. The results suggested that the amount of cellular-rich neointima was significantly lower following the GNR-HDL injection. Furthermore, the HDL treatment caused a reduction in the macrophage content in the injured artery (compared to the injection of GNRs alone), as demonstrated by both high-resolution CT images and immunostaining of the macrophage cell marker CD68.

Since, in the current study, GNP-HDLs were associated with a reduction in the LP-PLA2 release from macrophages, and also with macrophage accumulation in the injured site, GNP-HDL therapy may contribute to plaque stabilization and prevent ASVD complications. Although it has been reported that systemic HDL treatment reduced the atherosclerotic unstable plaque burden, this was achieved only by a series of intravenous infusions [6,7], and further research with clinical endpoints remains essential [53,54]. In addition to the therapeutic properties of HDL, it is an important natural molecule that can be modified for biomedical imaging purposes and the targeting of atherosclerotic macrophage-rich plaques [52]. An in vivo study in an atherosclerotic animal model showed that nanocrystal HDL was able to serve as a targeted contrast agent for CT scanning [16,55]. However, the current study is the first to show that GNR-HDL could serve as a detection tool for macrophage-rich plaques using DR measurement, and also as a potential therapeutic agent for the modification of inflammatorily active unstable plaques.

Both the in vitro and in vivo experiments in this study indicate that GNPs coated with HDL not only enabled the detection of macrophage-rich vascular lesions, but also inhibited the inflammatory response in these lesions. Although HDL is a well-known therapeutic target for ASVD, the novelty of our study is its use of GNRs as carriers for HDL, potentially enabling early detection of unstable atherosclerotic plaques using the DR simple method. In addition, we used the GNR-HDL to detect the unstable plaque using the simple, noninvasive and highly sensitive DR method, rather than the sophisticated imaging methods of MIR or CT. In conclusion, the combination of GNR-HDL presents a promising tool for the simultaneous detection and specific treatment of macrophage-rich atherosclerotic unstable plaques. Further investigation is required to underline the mechanisms by which HDL reduces LP-PLA2, and examine the involvement of other cytokines. These findings should be expanded and verified by using other atherosclerotic models.

## Figures and Tables

**Figure 1 diagnostics-12-00577-f001:**
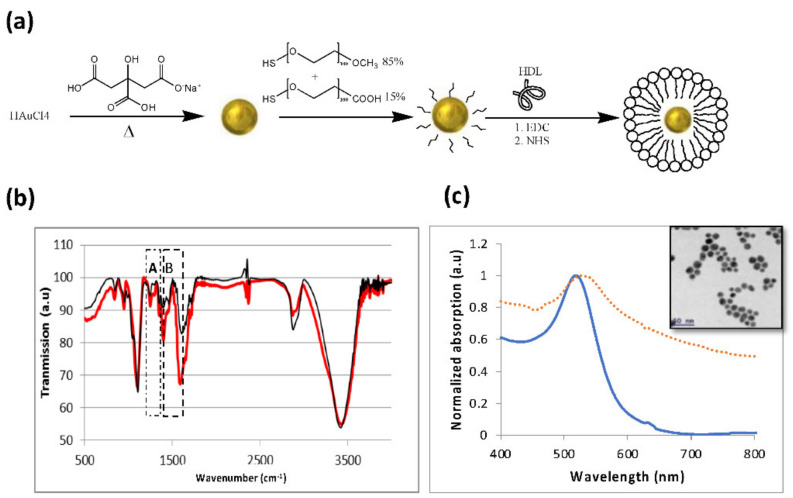
Characterization of GNSs and GNS-HDL. (**a**) Schematic diagram of the GNS synthesis process and the coating of GNS with m-PEG (85%) and COOH–PEG (15%), followed by covalent conjugation with HDL. (**b**) FT-IR spectra of GNSs (thin line) and GNS-HDL (bold line). The peaks around 950 cm^−1^ are the major proof of the formation of the C–N bond. (**c**) Optical properties of GNSs: ultraviolet-visible spectroscopy of bare GNSs (solid line) and HDL coated GNSs (dotted line), both presenting an absorption peak at 530 nm. Inset: TEM image of the 20 nm gold nanospheres.

**Figure 2 diagnostics-12-00577-f002:**
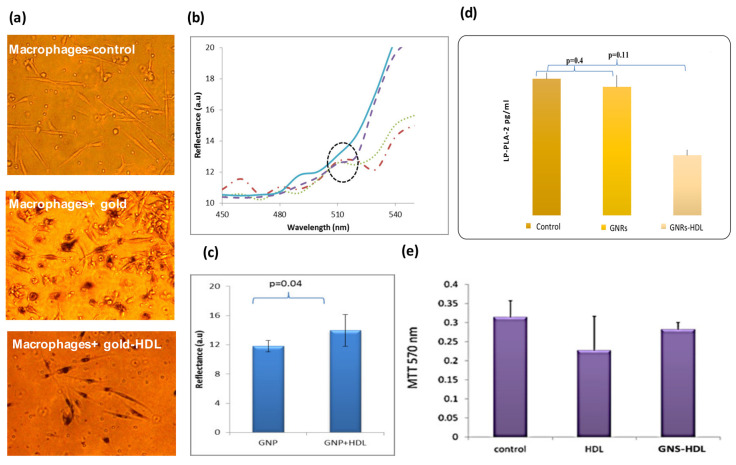
In vitro measurements of macrophage cell culture following incubation with GNSs and GNS-HDL. (**a**) GNSs uptake by macrophages captured by inverted microscopy. Nanoparticles appear as dark dots within cells due to light absorption by the particles. All experiments were performed with a magnification of 200×. (**b**) Reflectance intensity spectra of macrophages 48 h after their incubation with 0.1 mg/mL of GNSs were extracted from hyperspectral microscopy measurements. The reflectance spectra of macrophages with GNSs (dashed line) and with GNS + HDL (dotted and dashed-dotted lines) present an intensity peak at 530 nm, very similar to the absorption peak of the GNSs measured by the spectrophotometer (Figure 1c). The reflectance spectrum of the macrophages without GNSs (solid line) did not present a peak of around 530 nm. (**c**) Reflectance intensity at 530 nm was measured for *n* = 3 plates of macrophages with GNSs and macrophages with GNS + HDL. On average, higher intensity was measured for macrophages that were incubated with GNS + HDL, suggesting their superior uptake by the macrophages. (**d**) LP-PLA2 levels in macrophage cell culture media, using ELISA assay, after incubation with GNRs coated with HDL or with GNRs alone (*n* = 10). (**e**) MTT viable test of cell culture macrophages. Results show that both HDL and GNS-HDL were not toxic to the cells. For all experiments, *n* = 10.

**Figure 3 diagnostics-12-00577-f003:**
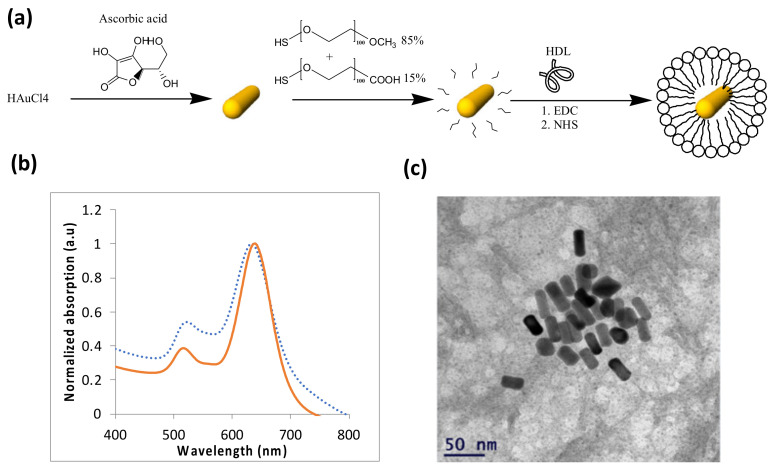
Characterization of GNRs. (**a**) Schematic diagram of the GNRs synthesis process and the coating of GNRs with m-PEG (85%) and COOH–PEG (15%), followed by covalent conjugation with HDL. (**b**) Optical properties of the GNRs: ultraviolet-visible spectroscopy of bare GNRs (solid line) and HDL coated GNRs (dotted line). (**c**) Transmission electron microscopy image of the GNRs. Average dimensions were 35 × 15 ± 2.2 nm (*n* = 10).

**Figure 4 diagnostics-12-00577-f004:**
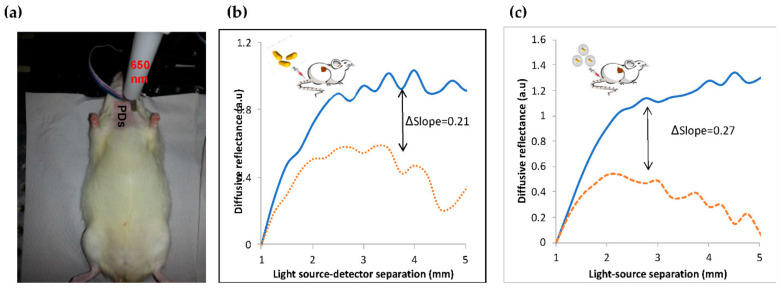
Noninvasive diffusion reflection measurements of injured rats’ carotids. (**a**) DR measurements of the rats’ carotids with a 650 nm laser diode illumination and four photodiodes (PDs) placed between 1 and 5 mm from the light source with a 1 mm separation. Representative DR curves of the healthy and injured carotids: (**b**) GNRs injection: DR profiles of injured arteries before (solid line) and 24 h after (dotted line) the GNRs injection. (**c**) GNR-HDL injection: Injured artery before (solid line) and 24 h post (dashed-dotted line) the GNRs injection. The Δslopes are due to the increase in the DR slope due to the GNRs injection, indicating that both GNRs and GNR-HDL injections increased the absorption of the carotids.

**Figure 5 diagnostics-12-00577-f005:**
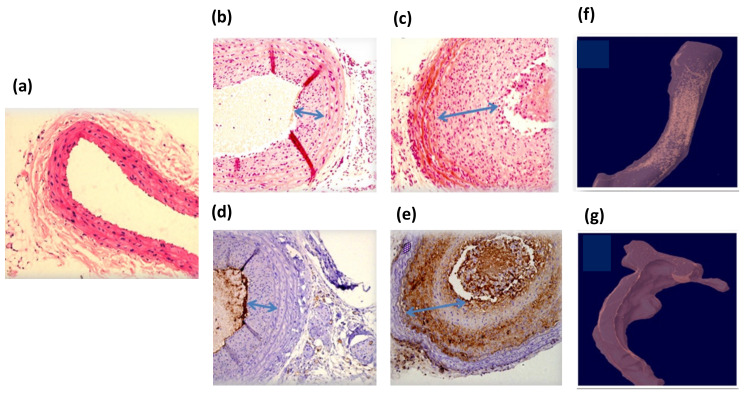
Ex vivo high-resolution CT scan and histology of rats’ injured arteries two weeks post GNRs injection. Panels (**a**–**c**): Hematoxylin and eosin (H and E) staining of normal carotid (**a**) and balloon-injured carotid artery 4 weeks post-injury and 2 weeks after GNR-HDL injection or GNRs injection (**b** and **c**, respectively). Panels (**d**,**e**): CD-68 immunostaining for macrophage accumulation in the injured carotid 4 weeks post-injury and 2 weeks after GNR-HDL injection or GNRs injection (**d** and **e**, respectively). No positive immunostaining was observed in the carotids treated with GNR-HDL, but a clear CD68 positive staining was observed in carotids treated with GNRs only. Magnifications are 40× in panel (**a**) and 100× in panels (**b**–**e**). (**f**) CT scan of the injured artery following GNRs injection. Gold is still apparent, indicating the presence of the macrophages within this artery. (**g**) CT scan of the injured artery following the GNR-HDL injection. The artery presented no GNRs accumulation, suggesting macrophage detachment from the injured site.

**Table 1 diagnostics-12-00577-t001:** Zeta potential and dynamic light scattering (DLS) size measurements (at 25 °C) of bare GNSs and HDL coated GNSs. The significant difference that was obtained (by zeta potential, DLS and UV-vis spectroscopy) following coating demonstrates the efficiency of the chemical coating.

Sample	Size, nm	Zeta, mV
GNSs	24.5 ± 0.3	−32.5 ± 0.6
GNS-HDL	44.9 ± 1.7	−23.9 ± 0.6

**Table 2 diagnostics-12-00577-t002:** Zeta potential and dynamic light scattering size measurements of bare GNRs and HDL coated GNRs. The significant difference that was obtained (by zeta potential, DLS and UV-vis spectroscopy) following coating demonstrates the efficiency of the chemical coating.

Same	Size, nm	Zeta, mV
GNR	50 ± 0.8	31.3 ± 0.6
GNR-HDL	91.22 ± 2.9	12.2 ± 0.6

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
