# Peer review of "Simultaneous Noninvasive Detection and Therapy of Atherosclerosis Using HDL Coated Gold Nanorods"

_diagnostics, 2022, doi:10.3390/diagnostics12030577_

Round 1

Reviewer 1 Report

The authors demonstrated noninvasive detection of atherosclerosis with HDL coated gold nanorods. The topic is very interesting and of great importance. The paper is well written with a lot of information. However, the figures of the manuscript should be revised carefully because they contained different fonts and overlappings. And the resolution of several figures are not high enough for readers to get the important information. Tables shoud not be included in the figure.

Author Response

We thank the reviewer for his comments. We have revised and improved the figures’ quality, making sure all fonts are identical and there is no overlapping between panels. We have also extracted the tables from the figures.

Reviewer 2 Report

Authors have demonstrated use of gold nanoparticles (GNPs) coated with high density lipoprotein (HDL) (GNP-HDL) for the  simultaneous detection and therapy of unstable plaques. They injected  the GNP-HDL on rats injured carotids, a  model for unstable plaques,to promote the reverse cholesterol transport (RCT) . GNP injection was enabled the noninvasive detection of the plaques  with Diffusion Reflection (DR) with a clear indication of the  accumulation of GNP-HDL particles at the injured site. They also tested their protocol ex vivo and  demonstrated the recovery of the injured artery treated with the GNP-HDL.  This is an interesting piece of  work that clearly demonstrated the applicability  of  GNP-HDL as a cost effective, simple and highly sensitive method for unstable plaque diagnosis and recovery.

The drafted manuscript is overall written well and the contents are presented nicely and systematically. However, the the manuscript needs a revision to address the following suggestions/ concerns.

  1. It would be great if authors could clearly emphasize the bio-compatibility and toxicity of the used GNPS.
  2. Authors are advised to provide a more general idea of GNPs used in, GNRs for contrast enhancement and pathology detection to get an idea for the wide range of readers.  Authors may find these these details from the articles : [1] https://doi.org/10.1088/2057-1976/2/5/055005 [2] https://doi.org/10.1166/jbn.2016.2297
  3. Authors can also make couple of statements regarding the performance of GNPs relative to silver nanoparticles.
  4.  GNPs can act as excellent contrast agents for the non-invasive imaging of GNPs embedded tissues optical coherence tomography (OCT) . Did authors perform imaging with such modalities  ? or panning in future ? 
  5. Can authors mention any restrictions in use such conjugated GNPs in human trials ? 

Author Response

We thank the reviewer for his comments. Please find below our attention to the referee suggestions:

1. Figure 2 in our manuscript shows the in vitro tests that have been done to track the macrophages cells viability following the uptake of the GNPs and GNPs-HDL. Figure 2(a) shows microscopy images of the cells following the GNPs uptake, and the high viability of the cells is well seen. In Figure 2(e) we show the MTT test results for macrophages following the uptake of the GNPs-HDL, as mentioned in text: “Cell viability was also evaluated using an MTT test (Figure 2(e)), indicating that the HDL is not toxic to the cells and does not have a negative effect on macrophage viability”. Following the reviewer’s comment, we have added a sentence that emphasize the bio-combability of the used GNPs:

“The cellular uptake of the GNS and GNS-HDL particles is observed as dark dots within the cells. Similar to results from our previous study, the macrophages with the GNPs show high viability presenting healthy cells morphology.”

2. In order to emphasize the use of GNRs in biomedical imaging, we have added the following paragraph to the introduction:

GNRs are well known contrast agents in biomedical imaging, showing high performance in optical coherence tomography (OCT), magnetic resonance imaging (MRI), coherence tomography (CT), terahertz spectroscopy, and more. All these methods are robust and widely used in clinical and/or in research applications, but suggest expensive and sophisticated equipment and do not provide a way to distinguish between stable and unstable AS plaques.

We have also added a citation, within this paragraph, of the paper proposed by the reviewer:

https://doi.org/10.1088/2057-1976/2/5/055005

3. We thank the reviewer for this comment. There are several reasons that we have preferred Au-NPs over Ag-NPs: i) Au is an inherent metal (due to, e.g., their very low Ksp value) which has been shown to be non-toxic to living cells. In addition, Au- NPs show higher SPR which make them better contrast agents in compared to the Ag-NPs (see, e.g., Comparative study of one pot synthesis of PEGylated gold and silver nanoparticles for imaging and radiosensitization of oral cancers, Shameer et.al., 2022 or Pietroal., Gold and Silver Nanoparticles for Applications in Theranostics, 2016).

We preferred not to mention it in our work since we did not test Ag-NPs in our applications.

4. We thank the reviewer for providing us with the idea of using GNPs as contrast agents in OCT. We sure hope to use our GNRs-HDL in OCT in the  future.

5. To the best of our knowledge, among all types of conjugated GNPs, PEGylated AuNPs are the only type of GNPs that were FDA approved for human trials. We sure hope that the GNPs-HDL will be soon approved for the treatment and tracking of atherosclerosis in human.